# The Immediate Effects of Muscle Energy Technique in Chronic Low Back Pain Patients with Functional Leg Length Discrepancy: A Randomized and Placebo-Controlled Trial

**DOI:** 10.3390/healthcare12010053

**Published:** 2023-12-26

**Authors:** Jung-Dae Yoon, Jin-Hwa Jung, Hwi-Young Cho, Ho-Jin Shin

**Affiliations:** 1Department of Health Science, Gachon University Graduate School, Incheon 21936, Republic of Korea; zzungdae89@gmail.com; 2Barum Pilates and Rehabilitation Center, Incheon 22014, Republic of Korea; 3Department of Occupational Therapy, Semyung University, Jecheon 27136, Republic of Korea; otsalt@semyung.ac.kr; 4Department of Physical Therapy, Gachon University, Incheon 21936, Republic of Korea

**Keywords:** muscle energy technique, pelvic alignment, low back pain

## Abstract

This study was conducted to determine the effect of muscle energy technique (MET) on pelvic alignment, leg length, pain, and fatigue in chronic low back pain (CLBP) patients with leg length discrepancy (LLD). Forty-two CLBP patients with LLD volunteered to participate and were randomly assigned to the MET group (*n* = 21) and placebo group (*n* = 21). The intervention group performed three METs with 5 s of isometric contraction and 30 s of rest once, and the placebo group performed three times the placebo-MET, maintaining the same posture as the MET group without muscle isometric contraction. X-ray equipment, tape measure, and visual analog scale were used to evaluate pelvic alignment, leg length, pain, and fatigue before and after each intervention. In comparison pre- and postintervention, only the MET group showed significant changes in pelvic alignment, leg length, pain, and fatigue (*p* < 0.05). In comparison between groups, there were significant differences in all variables (pelvic alignment, leg length, pain, fatigue) (*p* < 0.05). The results of this study confirmed the therapeutic effect of MET for improving pelvic alignment, functional LLD, pain, and fatigue in CLBP patients with functional LLD. Future research is needed to evaluate the long-term effect on more chronic low back pain patients.

## 1. Introduction

Chronic low back pain (CLBP) is a common health problem that affects people of all ages worldwide, affecting more than 80% of the working-age population at least once in their lifetime [1,2,3]. In particular, CLBP has the highest prevalence among work-related musculoskeletal diseases in office workers and is one of the leading causes of functional limitations in work and daily life [4]. It is also recognized as a socioeconomically significant problem, as it is associated with high healthcare costs [5,6]. 

Multifactorial causes have been implicated in developing low back pain (LBP), including disc degeneration, spinal stenosis, foraminal stenosis, facet joint degeneration, and damage to muscles, fascia, and ligament [7]. In addition, pelvic asymmetry is also a potential contributor to the development of LBP and is utilized by chiropractors and physiotherapists in diagnosing and treating low back pain in clinical practice [8,9,10]. Pelvic asymmetry increases the leg length discrepancy (LLD) between the lower extremities [11]. It can lead to an imbalance in the muscles around the pelvis, and these changes in the body can cause LBP. LLD is categorized into structural LLD (SLLD) and functional LLD (FLLD), of which FLLD is associated with pelvic asymmetry [11]. LLD, which affects up to 70% of the population, is accompanied by pelvic asymmetry, which can cause and increase stress and strain on structures around the lower back (muscles, ligaments, joint capsule, intervertebral disc, etc.), leading to LBP [12,13,14,15,16].

Therapeutic exercise, stretching, manual therapy, and tapping are used in clinical practice to intervene in LBP caused by pelvic asymmetry or muscle imbalance [17,18,19,20]. Among them, the muscle energy technique (MET) is one of the various manual therapy techniques used worldwide. MET is used when a healthcare provider instructs a patient to perform a movement and induces voluntary muscle contraction to lengthen shortened or contracted muscles, strengthen physiologically weakened muscles, reduce localized edema, improve joint mobility, and reduce pain [21,22,23]. MET is also an effective treatment technique for managing lumbopelvic pain and has been reported to restore muscle imbalances in the lower back and pelvic region, such as pelvic asymmetry [20]. In addition, previous studies reported that MET positively improved pelvic asymmetry and LLD in patients with sacroiliac joint dysfunction, as well as mediating pain [24,25,26]. Although these studies have shown positive effects of MET on various symptoms of LBP, they have used traditional physical therapy or therapeutic interventions along with MET. While this study demonstrated the effectiveness of MET as an adjunct to conventional physical therapy, the effectiveness of MET alone is unclear. In clinical practice, MET is often applied only to patients with low back pain, so it is necessary to clarify the evidence on the effectiveness of MET alone. In addition, previous studies have been conducted as single case studies or have small sample sizes, which limits the generalizability of the results. In addition, even though MET is effectively used in clinical practice to correct patients’ posture, no studies have used tools that can objectively measure structural changes, such as X-rays, to demonstrate this effect.

Therefore, this study aims to determine the effect of MET on pelvic alignment, LLD, pain, and fatigue in CLBP patients with LLD. In particular, it is the first attempt to identify objective structural changes using equipment such as X-rays and to determine the relationship between structural changes and changes in pain and fatigue in patients with CLBP.

## 2. Materials and Methods

### 2.1. Participants

In this study, 42 participants meeting the inclusion criteria were recruited among 60 adults aged 20 to 40 diagnosed with chronic LBP at Hospital I in Incheon. The inclusion criteria for the participants are as follows. (1) diagnosed with chronic LBP at least six months ago, (2) with FLLD of 20 mm or more, (3) with one side of pelvis raised (up-slip), and (4) have not previously received MET before. The exclusion criteria for the subjects are as follows. (1) with orthopedic or neurological spinal diseases; (2) with acute LBP; (3) with chronic pain of lower extremity; (4) underwent surgery or suffered multiple traumas more than six months ago; (5) congenital hip dysplasia; (6) dysfunctional sacroiliac (SI) joint; and (7) with SLLD.

The participants received a thorough explanation of the experimental process of this study and signed the consent form for participation. This study was approved by the Gachon University Bioethics Review Committee (1044396-201802-HR-055-01) and registered with the Clinical Research Information Service in compliance with World Health Organization regulations (KCT0006430).

### 2.2. Experimental Procedure

This study was conducted for three months, from March 2018 to May 2018, and was designed as a randomized controlled trial (RCT). Participants were randomly assigned into two groups by using permuted block randomization (block size 4). For randomization, the Random allocation software program (version 2.0, Informer Technologies, Inc., Isfahan, Iran) was used [27], and this experiment was conducted as a single session.

Before the experiment, the researcher measured the demographic characteristics of the participants. The researcher assessed pelvic alignment, leg length, pain, and fatigue levels before administering the intervention. Subsequently, a single session of the intervention was applied in both groups. After the intervention, a post-test was conducted under the same conditions as the pre-test. The rest time between intervention and measurement was set to 5 min to prevent fatigue caused by the intervention. 

A physical therapist with more than four years of clinical experience performed the MET intervention. Measurements were performed under blinded conditions by two physical therapists with a master’s degree and at least three years of clinical experience. The experimental procedure of this study is presented in Figure 1.

### 2.3. Intervention

#### 2.3.1. MET Group

Participants were instructed to lie down on the table with their ankles extended to the edge of the table. The therapist stood at the end of the table and fastened one of the participant’s feet to his thigh. In this position, the therapist held the ankle where MET was to be applied with both hands and placed the SI joint in a loose-packed position and the hip joint in a closed-packed position (hip joint 10–15° abduction and complete internal rotation). Then, while the participant held a deep breath, the physical therapist pulled the lower limb along the long axis of the leg to a point where the participant could no longer pull (movement barrier). Finally, the participants performed an isometric contraction for 5 s against the therapist’s distal pulling force in the long axis direction. The participant is instructed to perform a 5 s isometric contraction at approximately 25% maximal effort. A total of 3 sets were performed, and the rest time between sets was set at 30 s. After completing these three sets, the participant was instructed to take a deep breath 3–4 times and clear their throat. Simultaneously clearing the throat, the therapist momentarily pulled participant’s leg in a caudally direction (Figure 2) [21].

#### 2.3.2. Placebo Group

Participants were instructed to lie on the table in the same position as those receiving the MET intervention. After aligning the SI joint and hip joint in the same way as in the MET group, the therapist held the participant’s ankle for 5 s without providing resistance to prevent the participant’s pelvic muscle contraction. A total of 3 sets were performed, and the rest time between sets was set at 30 s. After completing three sets, the participant was instructed to take a deep breath 3–4 times and clear their throat.

### 2.4. Outcome Measurements

This experiment used radiography readings, a tape measure, and a visual analog scale to measure pelvic alignment, leg length, pain, and fatigue.

#### 2.4.1. Pelvic Alignment

In this study, an X-ray beam limiting device (AccuRay 525R, Dong Kang Medical Systems, Seoul, Republic of Korea) was used to measure pelvic alignment, and all radiographs were stored in a picture archiving and communication system (PACS, Agfa HealthCare, Mortsel, Belgium). The captured images are presented digitally, and the PACS system was used to measure the vertical distance and left and right tilt angles of the pelvis. For consistent measurement, eight landmarks (horizontal line, HL; the highest point of iliac crest, HI; the highest point of femoral head, HH; bi-ischial tuberosity, BI; anterior superior iliac spine, ASIS) were marked on the radiographic image of the pelvis (Figure 3).

Based on the displayed landmarks, the distance difference between horizontal lines (HL-HI, HL-HH, HL-BI) and the inclination angle of the left and right ASIS were collected as data [28,29,30]. Radiographs and data collection were performed by a radiologist with more than four years of experience. Radiographs have high intra-rater reliability (ICC = 0.90) and inter-rater reliability (ICC = 0.83) for pelvic measurements [29]. 

Measurements were conducted with the subject standing upright [28]. To minimize differences in measurements between subjects and to standardize the measurement method, the researchers instructed the subjects to stand barefoot with their feet shoulder-width apart during the measurement. The researchers marked the position of the subjects’ feet with masking tape to ensure that the shape of the feet was the same before and after the intervention. They were instructed to keep their head fixed straight ahead and look straight ahead. Measurements were repeated three times, and the average value was used as data.

#### 2.4.2. Leg Length

In this study, leg length measurement was performed using a tape measure. Leg length was measured by a physical therapist with more than four years of clinical experience. Leg length measurements were performed with the participants lying supine on a table. Leg length was measured by structural leg length and functional leg length. The structural leg length test refers to the distance between ASIS and medial malleolus. The functional leg length lies at the bottom of the navel and medial malleolus. Measurements were repeated three times, and the average value was used as data. The participant walked in place ten times between each measurement [31]. The tape measure method for measuring leg length demonstrates high validity and intra-rater reliability (ICC 0.990 and 0.985) and inter-rater reliability (ICC = 0.991) [32].

#### 2.4.3. Pain and Fatigue

The intensity of pain and fatigue was measured using a visual analog scale (VAS). All participants indicated the intensity of their subjective pain or fatigue on a 100 mm line. All participants rated their subjective pain or fatigue on a 100 mm line. The researcher explained to the subjects that 0 was the absence of pain and 100 was the worst pain the individual could think of. VAS is an evaluation tool with very high intra-rater reliability (r = 0.99) and inter-rater reliability (r = 1.00) [33].

### 2.5. Data Analysis

The sample size was calculated using the G*Power software (G* Power ver. 3.1.9.2; University of Kiel, Aichach, Germany). The sample size was calculated by selecting ANOVA: repeated measures, within-between interaction of F test, and entered an effect size f = 0.25, α error prob = 0.05, power = 0.8, number of groups = 2, and number of measurements = 2 into the program, and a total of 34 persons were counted. Analysis of the collected data was performed using SPSS software (version 25.0 IBM corp., Armonk, NY, USA). The measured values of all items were calculated as mean and standard deviation. Normality was tested using the Shapiro–Wilk test. To analyze the general characteristics of the two groups, an Independent *t*-test was used when normality was satisfied, and the Mann–Whitney U-test was used when normality was not satisfied. Repeated-measure ANOVA was used to analyze the changes in variables (pelvic alignment, leg length, pain, and fatigue) between the groups over time. Independent *t*-tests were performed for comparison between the groups, and paired-*t* tests were performed for comparisons pre- and postintervention within the group. All statistical significance levels were set at α = 0.05.

The effect size was utilized Cohen’s f test. The partial eta squared (η^2^) effect size was classified as small (≥0.10), medium (≥0.25), and large (≥0.40).

## 3. Results

### 3.1. Characteristics of Participants

Forty-two subjects participated in this study, and all subjects did not complain of discomfort during the experiment. The demographic characteristics, pelvic alignment, leg length, pain, and fatigue of the subjects of this study are presented in Table 1, and there were no significant differences in variables between groups (*p* > 0.05).

### 3.2. Leg Length Discrepancy

There was a significant difference in comparison between groups (*p* < 0.05) (Table 2). In the within-group comparison, there was a significant difference only in the MET group (*p* < 0.05). Statistically significant time × group interaction was confirmed (*p* < 0.05).

### 3.3. Pelvic Alignment

There were significant differences in some variables of pelvic alignment (HL-HH, obliquity) between the group’s comparison (*p* < 0.05) (Table 2). In the within-group comparison, there was a significant difference in all variables only in the MET group (*p* < 0.05). Statistically significant time × group interaction was confirmed for all variables (*p* < 0.05).

### 3.4. Pain and Fatigue

Pain VAS was significantly different when compared between groups (Table 3). In the within-group comparison, there was a significant difference only in the MET group (*p* < 0.05). There was a significant difference in Fatigue VAS when compared between groups (*p* < 0.05). In the within-group comparison, there was a significant difference only in the MET group (*p* < 0.05). Statistically significant time × group interaction was confirmed for pain VAS and fatigue VAS (*p* < 0.05).

## 4. Discussion

The study found that MET effectively improved pelvic alignment, LLD, pain, and fatigue in CLBP patients with LLD, while the control group, which performed the same posture but did not induce muscle contraction, only improved pain. The detailed interpretation of the dependent variable results is as follows.

First, pelvic alignment and LLD showed significant improvement only in the MET group. The pelvis may develop postural malalignment due to the habit of maintaining poor posture or the muscle imbalance induced by such posture [34]. MET is an active muscle release technique that lengthens shortened muscles and improves the function of the musculoskeletal system [35]. MET also requires the subject to isometrically activate the muscle to perform the exercise, which activates the muscle’s proprioceptors, the Golgi tendon organ (GTO). After isometric muscle contraction, a muscle relaxation effect can be observed due to the response of GTO [36]. We speculate that MET applied to the muscles that caused the pelvic malalignment relaxed the pelvic muscles that were hypertonic or shortened, thereby restoring balance with the muscles on the other side and improving pelvic alignment. Consistent with our results, a previous study suggested that MET application improved pelvic alignment [20], and in addition to our results, they also found changes in leg length. In clinical practice, many physicians, physiotherapists, and chiropractors consider these factors in their diagnoses and interventions when applying various therapeutic techniques to patients. In addition, previous studies have reported that LLD and pelvic asymmetry are positively correlated and that increasing differences in LLD are associated with worsening pelvic asymmetry [37]. Thus, our findings provide the first evidence that MET can help restore not only pelvic symmetry but also functional leg length.

Second, MET has shown positive results in improving not only postural alignment, but also pain in the lower back region. Wilson et al. (2003) also showed a positive effect of 4 weeks of combined exercise and MET on improving back pain in patients with LBP [20]. The study also reported a more significant improvement in back pain in the MET group than in the exercise-only group, emphasizing the benefits of MET. This study also confirmed that MET is efficacious in improving pain in patients with LBP by showing changes beyond the MCID of the VAS (MCID 18–19 mm) [38]. Still, unlike the additive effect of MET in previous studies, this study showed the effect of MET alone on pain intervention. It is interesting to note that Wilson et al. (2003) [20] conducted an eight-session intervention for four weeks, while we conducted a single intervention but showed similar results. Both studies did not perform follow-up measures, so the duration and effectiveness of the intervention cannot be clearly determined, and it is possible that repeated interventions may have a more positive effect than a single intervention. However, this study demonstrates for the first time that MET can help improve postural alignment and pain and fatigue in LBP patients with a single intervention, and we propose the use of MET in the management of LBP in clinical practice. It is also speculated that improvements in pelvic alignment and LLD, the first-mentioned outcomes, may have contributed to the improvement in pain. Cohen (2005) suggested that abnormal motion or malalignment of the pelvis and SI joints increases the load on the joints in the pelvis and lumbar region, leading to low back pain [39]. In the present study, MET significantly improved pelvic alignment and LLD, and these changes may have improved pain by reducing the load on the joints in the lower back and pelvic region. Because joint malalignment increases the load on the joint capsule and can stimulate nociceptors, restoring joint alignment may be an important consideration for pain intervention. In contrast, however, Cibulka et al. (1988) suggested that changes in pelvis alignment were not associated with a reduction in low back pain [40]. Childs et al. (2004) also reported that improvements in positional malalignment of the pelvis were not associated with a patient’s self-reported pain level, whereas improvements in weight-bearing symmetry were associated with a reduction in pain [41]. However, these studies either used an inclinometer to measure unilateral innominate tilt or a static palpation test. Not only do the measurement methods used in these studies differ from ours, but the objective accuracy of the measurements also differs from ours, limiting the ability to compare results between studies directly. Interestingly, in previous studies, improvements in weight-bearing symmetry were significantly correlated with improvements in pain intensity in patients with low back pain. In addition, Levangie and Norkin (2011) reported that pelvic asymmetry caused by LLD shifts the body’s center of gravity toward the relatively shorter leg [42]. Therefore, although weight-bearing symmetry was not measured in this study, it is expected that improvements in pelvic symmetry would have led to symmetrical improvements in pelvic height, which in turn would have improved bilateral weight-bearing symmetry. In other words, the pelvic symmetry restored by the MET intervention may have contributed to improving pain. 

Third, MET significantly improved fatigue in LBP patients. As with pain, this is likely due to improved pelvic alignment and LLD. When pelvic malalignment and LLD are induced, compensations are triggered in other body parts to maintain proper posture, which involves constant activation of the muscles of the associated body parts. In turn, this leads to an increase in unnecessary energy expenditure related to posture maintenance and performance, which causes muscle fatigue. Therefore, the MET applied in this study improved patient fatigue by improving excessive muscle activity related to posture maintenance and performance through improving pelvic alignment and LLD. 

Limitations of this study include: First, this study examined the effect of a single application of MET, i.e., the short-term effect. As a result, the difference between the effects of long-term application of MET and short-term application needs to be clarified. Second, the interventionist in this study had completed multiple trainings in MET and orthopedic physical therapy and had extensive clinical experience. Therefore, it is limited to expect the same effect from clinicians with different experience levels. Third, the subjects performed MET in the supine position. Some previous studies applying MET have been performed in other positions, which limits the ability to compare results between studies [20,43]. Fourth, the duration of the intervention needed to be clarified. Therefore, to compensate for some of these limitations, further studies are needed to investigate the effectiveness of MET in CLBP patients, the duration of the effect, and the posture in which the intervention is applied.

## 5. Conclusions

The present study demonstrated that MET as a single intervention positively affected pelvic alignment, FLLD, pain, and fatigue in CLBP patients with FLLD. Based on this study’s results, the application of MET before exercise intervention may be one way to prevent poor movement patterns that may occur due to abnormal alignment during exercise intervention. Moreover, we suggest that MET be used in clinical and rehabilitation centers to improve pain and fatigue, as well as pelvic alignment and FLLD in CLBP patients with FLLD.

## Figures and Tables

**Figure 1 healthcare-12-00053-f001:**
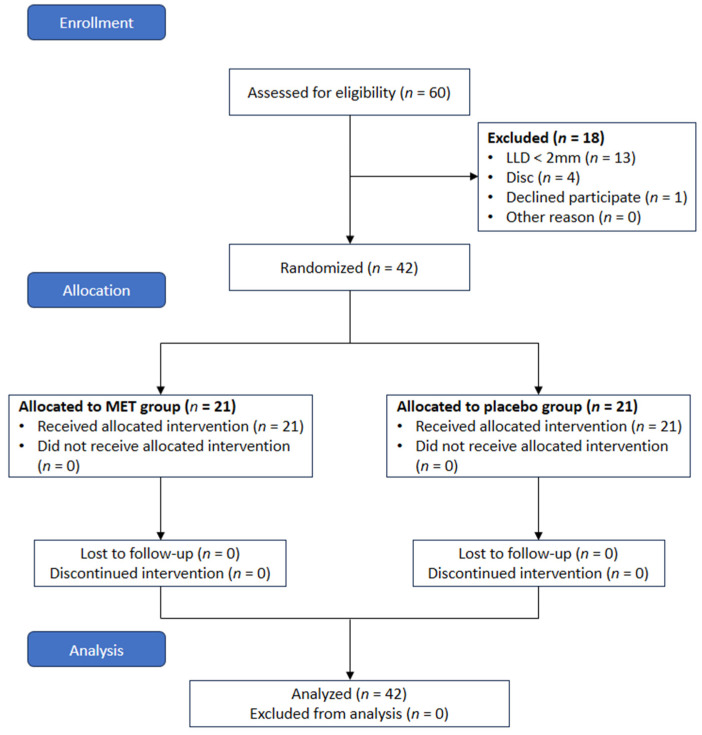
CONSORT flow diagram.

**Figure 2 healthcare-12-00053-f002:**
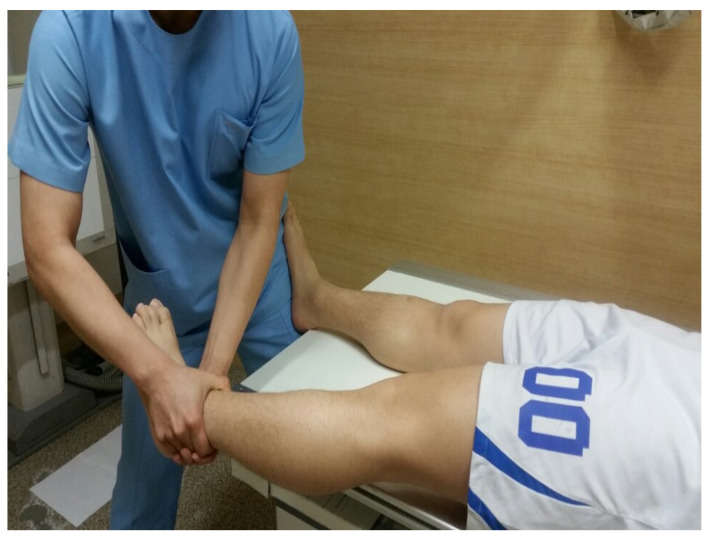
Muscle energy technique.

**Figure 3 healthcare-12-00053-f003:**
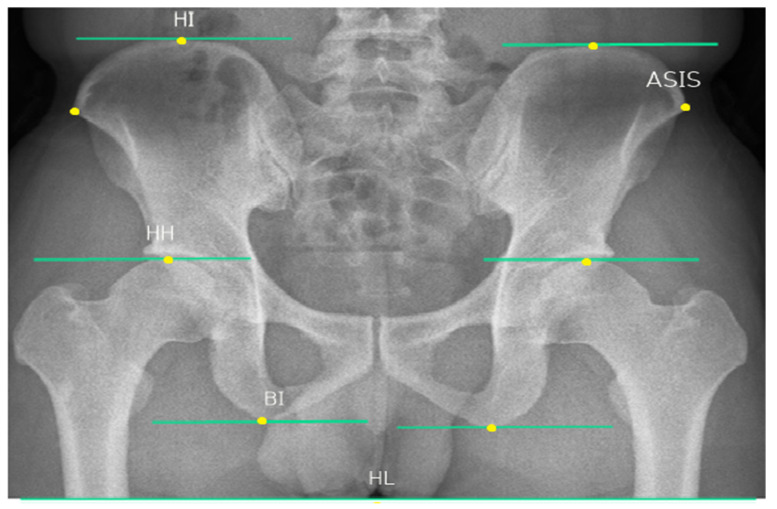
Pelvic landmark.

**Table 1 healthcare-12-00053-t001:** General characteristics of the participants.

Variables	MET Group	Placebo MET Group	*p*-Value
* Male/Female (*n*)	13/8	12/9	0.753 ^a^
Age (year)	31.14 ± 5.46	33.67 ± 5.66	0.149 ^b^
Height (cm)	168.33 ± 7.52	170.30 ± 7.69	0.408 ^b^
Weight (kg)	66.83 ± 12.35	65.52 ± 10.16	0.822 ^b^
BMI (kg/m^2^)	23.78 ± 5.08	22.79 ± 4.50	0.538 ^b^
TMM			
SLLD (mm)	0.24 ± 0.16	0.28 ± 0.18	0.434 ^b^
FLLD (mm)	2.29 ± 0.22	2.33 ± 0.28	0.552 ^b^
Radiography			
HL-HI (mm)	5.61 ± 4.59	5.74 ± 2.83	0.911 ^b^
HL-HH (mm)	3.96 ± 2.21	4.55 ± 2.10	0.588 ^b^
HL-BI (mm)	4.07 ± 2.17	3.84 ± 2.00	0.544 ^b^
Obliquity (°)	3.14 ± 1.19	2.70 ± 1.10	0.227 ^b^
VAS			
Pain (mm)	43.15 ± 15.99	45.62 ± 13.16	0.588 ^b^
Fatigue (mm)	47.81 ± 19.49	47.95 ± 22.26	0.982 ^b^

Values are expressed as mean ± standard deviation. * Values are expressed as numbers of participants. ^a^ Chi square test; ^b^ independent *t*-test. Abbreviation. BMI, body mass index; TMM, tape measure method; SLLD, structure leg length discrepancy; FLLD, functional leg length discrepancy; HL, horizontal line; HI, the highest point of iliac crest; HH, the highest point of femoral head; BI, bi-ischial tuberosity; VAS, visual analogue scale.

**Table 2 healthcare-12-00053-t002:** Comparison of pelvic alignment and leg length discrepancy.

Variables	Pre	Post	Source	F (df)	*p*	Partial η^2^
FLLD (mm)						
MET group	2.29 ± 0.22	0.45 ± 0.21 ^a,^*	Time	269.168 (1)	<0.001	0.87
Placebo group	2.33 ± 0.28	2.26 ± 0.58	Group	86.998 (1)	<0.001	0.69
			T*G	229.608 (1)	<0.001	0.85
HL-HI (mm)						
MET group	5.61 ± 4.59	3.83 ± 3.37 ^a^	Time	26.525 (1)	<0.001	0.40
Placebo group	5.74 ± 2.83	5.64 ± 2.59	Group	0.862 (1)	0.359	0.02
			T*G	20.969 (1)	<0.001	0.34
HL-HH (mm)						
MET group	3.96 ± 2.21	2.96 ± 1.85 ^a,^*	Time	8.243 (1)	0.007	0.17
Placebo group	4.55 ± 2.1	4.42 ± 2.04	Group	2.905 (1)	0.096	0.07
			T*G	4.783 (1)	0.035	0.11
HL-BI (mm)						
MET group	4.07 ± 2.17	2.79 ± 1.86 ^a^	Time	5.561 (1)	0.023	0.12
Placebo group	3.84 ± 2	3.79 ± 2.52	Group	0.409 (1)	0.526	0.01
			T*G	4.735 (1)	0.036	0.11
Obliquity (°)						
MET group	3.14 ± 1.19	1.61 ± 1.09 ^a,^*	Time	65.024 (1)	<0.001	0.62
Placebo group	2.70 ± 1.10	2.49 ± 1.11	Group	0.475 (1)	0.495	0.01
			T*G	37.660 (1)	<0.001	0.49

Data were analyzed with Repeated Measures ANOVA. Values are expressed as mean ± standard deviation. ^a^
*p* < 0.05 compared to pre within groups; * *p* < 0.05 compared to the CG in between the groups. Abbreviation. FLLD, functional leg length discrepancy; HL, horizontal line; HI, the highest point of iliac crest; HH, the highest point of femoral head; BI, biischial tuberosity. η^2^, small ≥ 0.10, medium ≥ 0.25, large ≥ 0.40.

**Table 3 healthcare-12-00053-t003:** Comparison of pain and fatigue.

Variables	Pre	Post	Source	F (df)	*p*	Partial η^2^
Pain (mm)						
MET group	43.15 ± 15.99	21.94 ± 14.34 ^a,^*	Time	159.099 (1)	<0.001	0.80
Placebo group	45.62 ± 13.16	44.38 ± 14.33	Group	8.078 (1)	0.007	0.17
			T*G	125.943 (1)	<0.001	0.76
Fatigue (mm)						
MET group	47.81 ± 19.49	22.76 ± 13.51 ^a,^*	Time	56.153 (1)	<0.001	0.58
Placebo group	47.95 ± 22.26	45.48 ± 23.25	Group	3.766 (1)	0.059	0.09
			T*G	37.764 (1)	<0.001	0.49

Data were analyzed with Repeated Measures ANOVA. Values are expressed as mean ± standard deviation. ^a^
*p* < 0.05 compared to pre within groups; * *p* < 0.05 compared to the CG in between the groups.

## Data Availability

The data presented in this study are available on request from the corresponding author.

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
