# Peer review of "The Immediate Effects of Muscle Energy Technique in Chronic Low Back Pain Patients with Functional Leg Length Discrepancy: A Randomized and Placebo-Controlled Trial"

_healthcare, 2023, doi:10.3390/healthcare12010053_

Round 1

Reviewer 1 Report

Comments and Suggestions for Authors

Thank you for the opportunity to review an interesting paper by Jung-Dae Yoon et al.:  The Effects of Muscle Energy Technique on Pelvic Alignment,  Leg Length, Pain and Fatigue in Chronic Low Back Pain Patients with Leg Length Discrepancy.

The authors present an evaluation of a single procedure for applying the muscle energization technique to patients with chronic back pain.

The paper contains a well-structured introduction, a description of the procedure and methodology, and concisely presented results. The strength of the paper is the methodology, i.e., a randomized study with a control group assessing subjective parameters (pain, confusion) as well as objective ones ( clinical and radiological measurements).

The discussion is correctly presented, containing rich and appropriate references to the literature. The authors accurately described the limitations of the paper.

Editorial errors

Line 50: client - better patient

Consort diagram- descriptions in boxes are incomprehensible:

The box " Allocated to MET group" should only have: "received allocated intervention,” and the box " Allocated to placebo" should only have "did not receive."

Titles and sub-headings are written differently and inconsistently, once in italics, once without italics, and not always in capital letters - please standardize.

 The title is too long and not catching. What about, e.g., The impact of muscle energy technique in chronic back pain patients - a randomized and placebo-controlled experiment?

Author Response

Dear Reviewers,

I would like to thank you for providing the opportunity to revise and resubmit the attached manuscript entitled “The Immediate Effects of Muscle Energy Technique in Chronic Low Back Pain Patients with Leg Length Discrepancy: A Randomized and Placebo-Controlled Trials” for publication in the healthcare.

We sincerely appreciate the editorial comments and reviewers’ helpful comments on our manuscript, which we ignored. We agreed with the points addressed by the Reviewers. We provide our responses to the Reviewers’ comments. Please review the attached files.

Reviewer 2 Report

Comments and Suggestions for Authors

Thank you for submitting your valuable manuscript to this journal.

This is a randomized controlled trial on 42 participants with chronic low back pain and leg length discrepancy (LLD). The present study aimed to investigate the acute effects of muscle energy technique (MET) on pelvic alignment, leg length, pain, and fatigue in the participants. X-ray equipment, tape measure, and visual analog scale were used to evaluate the outcome measures.

There are some points that need to be considered in the manuscript:

1-      Since the intervention is conducted in a single session and the outcomes of the study are measured before and after it, please revise the topic of the paper to “The acute effects of Muscle Energy …..”

2-      On the methods of the study, line 73 for the inclusion criterion number 2, please identify the type of LLD (functional or structural). Please exactly define superior pelvic dysfunction (upslips) for criterion number 3.

3-      On introduction section, line 58, using traditional physical therapy or other interventions along with MET don’t limit the effectiveness of MET alone in treating LBP! The might have additive effects, so you cannot determine the effects of each modality separately. Please revise this part or give some references for your claim. Also please revise line 60, for its English language and clarity.

4-      On figure 1, please correct the number of participants in each group who did not receive allocated intervention (n=21)! Also please revise the figure legend.

5-      On the intervention section for MET group, line 104, please explain the maneuver of placing the participants’ leg on the movement barrier.

6-      For line 105, please describe the intensity of the participants’ isometric contraction and the direction of the therapist’s force. Please also explain the thrust maneuver on line 108 in more detail.

7-      Line 133, please define the line HL-IC!

8-      The results of the study need significant revision! Functional leg length discrepancy (FLLD) which is an important inclusion criterion of the study, is missing in table 1 (General characteristics of the participants). In table 2, pre-treatment FLLD (mean +- SD) of the placebo group is 2.09+-0.41. This means some of the participants in this group had the FLLD of 1.27 centimeter and they should not had been included in the study!

9-      Please specify the measurement unit of FLLD in table 2!

10-  In table 2, the data of HL-HI are repeated for HL-HH! This is a dreadful inattention to the data of the manuscript!

11-  There are still big mistakes in the tables! Pre-treatment HL-BI of the MET group is 3.70+-2.96 in table 1 and 4.07+-2.96 in table 2!

12-  Since there are serious mistakes in reporting the data of the study, it is impossible to review the data analysis process of the whole project!

13-   In the conclusion of the study, how did you deduce that MET can effectively intervene in pelvic alignment associated with other musculoskeletal disorders and improve muscle imbalance and mobility?

Good luck.

Comments on the Quality of English Language

Moderate editing of English language required.

Author Response

(The authors gave the same response as above.)

Reviewer 3 Report

Comments and Suggestions for Authors

REVIEW GENERAL COMMENTS

The authors aimed to examine the effects of the Muscle Energy Technique on Pelvic Alignment, 2 Leg Length, Pain and Fatigue in Chronic Low Back Pain Patients 3 with Leg Length Discrepancy. This paper is well-written, and the topic is interesting. Below are my comments.

SPECIFIC COMMENTS

Abstract

It is written correctly. Gives highlights from each section of the paper.

To fix: Line 16 - It can be confusing when using MET twice in the same sentence. Replace MET here with the intervention group. 

Introduction

The authors provided a good background. The reasoning logically follows the pattern correctly: Known → Unknown → Research question/hypothesis. The gap in the literature to be filled is described.

To fix: Lines 39 - 40 - Please cite previous research to support the statement.

Methods

Overall, the methodology is clearly explained. However, a priori statistical power analysis was not performed to calculate the sample size to be recruited.

The assessment tools used are validated and the reliability of all assessment tests, however I would like to see ICC calculated The statistical techniques used are appropriate.

Results

The results are written correctly. The tables are explanatory.

To fix: -Write in the captions of tables 1, 2, and 3, the statistical test used (e.g. unpaired t-test etc)

Discussion

The discussions are clear and to the point. The limitations are described.

To fix: Lines 286-287 you make reference to studies, but you do not cite any. Please cite the studies that you are referring to. 

Conclusions

The authors' conclusions are justified. The take-home message is clear.

To add: -Please help better readers understand how this paper is different from others already published.

Author Response

(The authors gave the same response as above.)

Reviewer 4 Report

Comments and Suggestions for Authors

Please see attachment file

Author Response

(The authors gave the same response as above.)

Round 2

Reviewer 2 Report

Comments and Suggestions for Authors

Thank you for your responses to the feedback and for submitting the revised version.

There are still some points that need more explanation in the manuscript:

1-      In radiographic landmarks, where did you locate the horizontal line (HL)? In figure 3 the HL is at the bottom of the X-ray, but as you reported the data in table 1 and 2, for example in MET group, HL-HH is 3.96 mm and HL-BI is 4.07 mm. Please define the exact location of the HL in the manuscript.

2-      Another important point in your data analysis is to show that whether the data are normally distributed. Since you reported the data as mean (+/- SD), you have to prove the normally distributed data. In normally distributed data, 95% of the data lie between the mean and 2 SD below and above it. It seems that your data are not normally distributed after the intervention! The FLLD in MET group which is the main outcome measure in your study is 0.45 + - 0.31 mm. This means that 95% of your cases have the values between -0.17 to 1.07 mm and your intervention has reversed the leg length of some patients to the other side. This point is more evident in pain and fatigue after the intervention. For example the pain in MET group after the intervention is 21.94 + - 14.34 mm. This means that 95% of your cases have the pain values between -6.74 to 50.62 mm and we know that the negative pain score is meaningless because the scores are between 0 – 100. So you have to use the non-parametric tests for data analysis in your results and not the repeated measure ANOVA!

3-      Since one important inclusion criterion in your study is FLLD of 20 mm or more, Please add the word “functional” in the title and conclusion of the study. The title of your paper would be “The Immediate Effects of Muscle Energy Technique in Chronic Low Back Pain Patients with Functional Leg Length Discrepancy: A Randomized Placebo-Controlled Trial”.

Good luck.

Comments on the Quality of English Language

Minor editing of the English language is required.

Author Response

Dear Reviewer,

I would like to thank you for providing the opportunity to revise and resubmit the attached manuscript entitled “The Immediate Effects of Muscle Energy Technique in Chronic Low Back Pain Patients with Leg Length Discrepancy: A Randomized and Placebo-Controlled Trials” for publication in the healthcare.

We sincerely appreciate the reviewer’ helpful comments on our manuscript, which we ignored. We agreed with the points addressed by the Reviewer. We provide our responses to the Reviewer’ comments. Please review the attached files.

Reviewer 4 Report

Comments and Suggestions for Authors

The manuscript became better through revisions.

Author Response

All the authors are deeply grateful for your kind words.